# ProtoSnap:
# Prototype Alignment for Cuneiform Signs

Rachel Mikulinsky[1]*    Morris Alper[1]*    Shai Gordin[2]
Enrique Jimenez[3]    Yoram Cohen[1]    Hadar Averbuch-Elor[1,4]

[1]Tel Aviv University    [2]Ariel University    [3]LMU    [4]Cornell University

## Abstract

The cuneiform writing system served as the medium for transmitting knowledge in the ancient Near East for a period of over three thousand years. Cuneiform signs have a complex internal structure which is the subject of expert paleographic analysis, as variations in sign shapes bear witness to historical developments and transmission of writing and culture over time. However, prior automated techniques mostly treat sign types as categorical and do not explicitly model their highly varied internal configurations. In this work, we present an unsupervised approach for recovering the fine-grained internal configuration of cuneiform signs by leveraging powerful generative models and the appearance and structure of prototype font images as priors. Our approach, *ProtoSnap*, enforces structural consistency on matches found with deep image features to estimate the diverse configurations of cuneiform characters, snapping a skeleton-based template to photographed cuneiform signs. We provide a new benchmark of expert annotations and evaluate our method on this task. Our evaluation shows that our approach succeeds in aligning prototype skeletons to a wide variety of cuneiform signs. Moreover, we show that conditioning on structures produced by our method allows for generating synthetic data with correct structural configurations, significantly boosting the performance of cuneiform sign recognition beyond existing techniques, in particular over rare signs. Our code, data, and trained models are available at the project page: https://tau-vailab.github.io/ProtoSnap/

## 1 Introduction

The earliest forms of decipherable scripts date back to the late 4th millennium BCE, with the invention of the cuneiform writing system in ancient Mesopotamia, which came to be used for a number of historically significant ancient languages such as Sumerian and Akkadian (Radner and Robson, 2011; Streck, 2021). Cuneiform signs have complex internal structures which varied significantly across the eras, cultures, and geographic regions among which cuneiform writing was used. The study of these variations is part of a field called *paleography*, which is crucial for understanding the historical context of attested writing (Biggs, 1973; Homburg, 2021). However, while computational methods show promise for aiding experts in analyzing cuneiform texts (Bogacz and Mara, 2022), they are challenged by the vast variety of complex sign variants and their visual nature: Represented as wedge-shaped imprints in clay tablets which have often sustained physical damage, cuneiform appears as shadows on a non-uniform clay surface which may even be difficult for human experts to identify under non-optimal lighting conditions (Taylor, 2015).

Prior work has focused on digitization of cuneiform tablets at a coarse resolution, localizing and classifying signs from photographs of whole tablets (Dencker et al., 2020; Stötzner et al., 2023a). However, these methods treat sign types as categorical while neglecting sign-internal configurations of strokes in each character, which provides crucial information for identifying rare signs and distinguishing between sign variants. In this work, we aim to recover the fine-grained internal configuration of real cuneiform signs, given coarse-grained categorical information as input. In

---

*Equal contribution

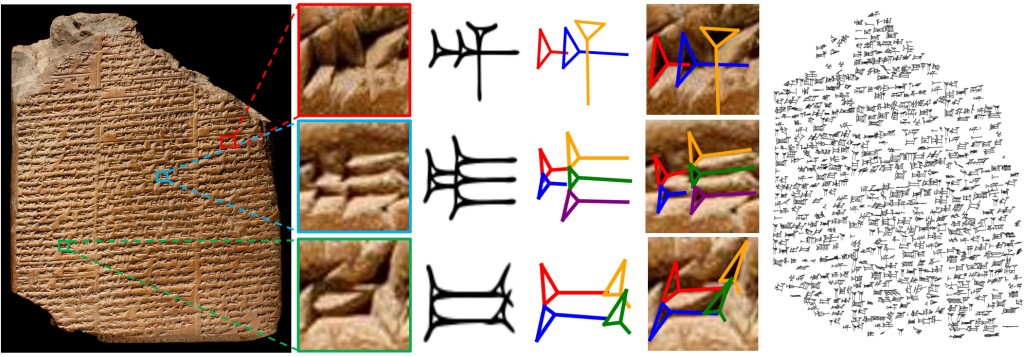

Real Cuneiform Tablet    Cuneiform sign   Prototype (font+skeleton)   Our output   Application: hand copy generation

Figure 1: *ProtoSnap* applied to a full tablet by cropping each sign using existing bounding boxes (such as those depicted in unique colors), and matching prototypes of the signs (illustrated in the center). Our technique "snaps" the skeletons of the prototypes to the target images depicting real cuneiform signs. These aligned results can be used to produce an automatic digital *hand copy* (right). We also show that our approach can be used to boost performance of cuneiform sign recognition.

particular, our method is provided with a prototype image and its associated skeleton indicating the canonical structure of a sign, and aligns this structure to a target image depicting a corresponding real cuneiform sign. As illustrated in Figure 1, our technique is analogous to the laborious *hand copies* produced by expert Assyriologists; when applied to a tablet with existing character-level annotations, this outputs the outlines of signs in the style of the original document. Furthermore, we show that these aligned skeletons may be used to boost optical character recognition (OCR) performance, by training a generative model with structural conditioning as detailed below.

To this aim, we present *ProtoSnap*, an *unsupervised* approach leveraging deep diffusion features to snap a skeleton-based prototype to a target cuneiform sign, revealing its structure without requiring any labelled examples of real photographed signs. By using a fine-tuned generative model as a prior on the appearance of cuneiform images, and enforcing global and local consistency, we are able to localize the constituent strokes in real cuneiform images. We make use of the key insight that pairwise similarities between regions of the prototype and target images encode information about both coarse global alignment and fine-grained local deviations of each stroke from its canonical pose. Our technique distills this information with a multi-stage process performing global alignment followed by local refinement of stroke positions.

We provide a new benchmark of expert-annotated photographed signs for evaluation, and show that our system succeeds at identifying their internal structures, significantly outperforming generic correspondence matching techniques. We also show the downstream utility of *ProtoSnap* for automatic digitization of cuneiform texts, by using aligned prototype skeletons as a condition for a generative model to produce structurally-correct synthetic data to train cuneiform OCR. Our results show that this achieves state-of-the-art results on cuneiform sign recognition, particularly enhancing performance on rare signs where naive synthetic data generation struggles to produce instances of the correct sign.

Stated explicitly, our key contribution are:

- *ProtoSnap*, an unsupervised prototype alignment method capturing the structure of photographed cuneiform signs.

- A novel benchmark with expert annotations, and results showing that our method outperforms generic correspondence methods at this task.

- State-of-the-art OCR results for cuneiform when using synthetic data produced using our method's alignments for conditional generation.

## 2 RELATED WORKS

**Machine learning (ML) for cuneiform.** Ancient texts provide a window into our history, but their decipherment and interpretation require painstaking work and expert knowledge of esoteric languages, complex writing systems, and historical context which serve as a barrier to their translation and analysis on scale. Due to the high societal value of these tasks and the scarcity of expert knowledge and time, machine learning promises to provide an invaluable aid for understanding the ancient world. In the context of a number of works on ML applied to diverse ancient inscriptions (Hassner et al., 2013a; Assael et al., 2019; Yin et al., 2019; Huang et al., 2019; Luo et al., 2021; Hayon et al., 2024), the cuneiform script poses particular challenges. These include the nature of the physical writing media (indentations in textured and often damaged clay under various lighting conditions), and the diverse nature of cuneiform signs which changed over thousands of years of use in vast geographical regions.

Various approaches have been applied to modeling cuneiform signs for the purpose of downstream tasks such as optical character recognition. Some works apply recognition directly to image data (Bogacz et al., 2017; Dencker et al., 2020; Stötzner et al., 2023a), while others have treated cuneiform as structured graphs to recognize signs based on their internal configurations (Kriege et al., 2018; Chen et al., 2024). A handful of works have explicitly modeled cuneiform signs as compositions of wedges, though these mainly focus on segmentation from 3D meshes or localizing strokes on the bounding box level (Bogacz and Mara, 2022; Stötzner et al., 2023b; Hamplová et al., 2024). By contrast, our approach provides a *pixel-aligned* skeleton indicating the relative positions, sizes, and orientations of the strokes, and unlike prior works we operate exclusively on 2D photographs of cuneiform signs and without any strong supervision.

**Skeleton-based template alignment.** Our approach to inferring the configuration of cuneiform images as compositions of strokes by aligning a skeleton-based template bears partial similarity to various existing methods that typically operate over generic natural images.

Our method resembles template matching in that we use a template image (our font prototype) and search for relevant regions in the target image. While earlier approaches used naive comparisons of image intensities or low-level features when sliding the template across the target image (Ben-Arie and Rao, 1993; Cole et al., 2004; Kim et al., 2011), this is not robust to complex relations between the template and target. To handle these challenges, more recent works have adopted deep image features along with modeling complex non-rigid deformations (Oron et al., 2017; Talmi et al., 2017; Cheng et al., 2019; Gao and Spratling, 2022). Our method similarly searches for matches to our template using deep features and allowing for deformations, although we differ from conventional template matching in explicitly using the skeletonized graph structure of the template and matching each of its constituent strokes separately.

We also note similarity to pose estimation methods, as we infer the structure of a sign by localizing keypoints. Pose estimation methods align a graph of keypoints to an image, most commonly applied to a single category such as humans (Fang et al., 2022; Zheng et al., 2023), animals (Li and Lee, 2021; Yang et al., 2022), or vehicles (Reddy et al., 2018; López et al., 2019), where the same fixed graph applies to all instances. However, in our setting the number and connectivity of keypoints depends on the sign type under consideration. In this respect our method resembles category-agnostic pose estimation methods (Xu et al., 2022; Hirschorn and Avidan, 2023), though our input also includes a template image rather than only using an abstract graph.

In the context of images depicting text, a number of works address the problem of text spotting, which searches for matches to a given visual text representation in images (He et al., 2018; Huang et al., 2022; Ye et al., 2023). This may include alignment or dewarping of detected text, but does not typically explicitly leverage the internal shape of symbols as in our method. We also note several works performing transcript alignment using dense correspondence methods, which align visual text but do not explicitly handle character-internal structure (Hassner et al., 2013b; 2016).

## 3 THE CUNEIFORM WRITING SYSTEM

Cuneiform, one of the earliest known writing systems, was a logo-syllabic script indicating both units of sound and meaning with signs. Unlike the Latin alphabet which uses less than thirty basic

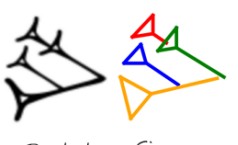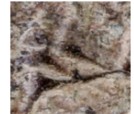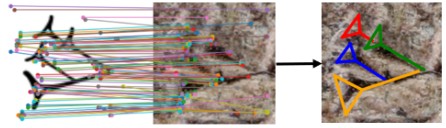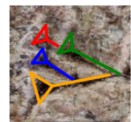

Prototype Sign
(Font + Skeleton)

Target
Image

Best-buddy Correspondences for
Global Alignment

Local
Refinement

Figure 2: **Method Overview**. Given a prototype image with annotated skeleton and a target image of a real cuneiform sign, *ProtoSnap* first extracts *best-buddy* correspondences from deep diffusion features (extracted with our fine-tuned SD-𒀭 model), globally aligning the target image to the skeleton of the prototype. Our method then "snaps" the individual strokes into place with a local refinement stage by optimizing a per-stroke transform.

letters to indicate sounds, cuneiform signs numbered upwards of one thousand unique types, which varied dramatically in their realizations across eras, languages, and geographical regions (Walker, 1987). See, for instance, the two right-most examples in Figure 5, both variants of the same sign AN from different eras. Scholars have attempted to collate lists of known signs and their variants (Labat and Malbran-Labat, 1988; Borger, 2003), and canonical representations of the most common variants for different time periods have been encoded in digital fonts.

Cuneiform was written on clay tablets by scribes using a stylus to create wedge-shaped impressions, also known as *strokes*, which combine to form signs. Scribes used styli with triangular edges to create impressions on the moist clay in three possible directions: horizontal, vertical, or oblique (Cammarosano, 2014; Cammarosano et al., 2014). Various encoding schemes proposed to encode these strokes digitally (Bogacz and Mara, 2022); we adopt the four-keypoint scheme, treating all strokes as being composed of a triangular head indicated with three keypoints and a fourth keypoint indicating the stroke's tail; see Figure 2 (second to left) for an example. We refer to the graph of these keypoints and the edges connecting them as a sign's *skeleton*. Our method's prototype input consists of both a rasterized font image along with its skeleton encoding the configuration of its strokes; we collect these skeletons via manual annotation as described in the appendix.

## 4 METHOD

Given a prototype consisting of a font image annotated with a skeleton composed of strokes and a target image of a real cuneiform sign, our system aligns the prototype skeleton to match the sign structure in the target image. An overview of this process is shown in Figure 2.

Our system consists of three steps: First, we calculate a semantically-adapted 4D similarity volume which encodes the pairwise feature similarities of each pair of regions in the two images. This similarity volume uses diffusion features to encapsulate the complex geometry and semantics of cuneiform images. We then calculate a global alignment between the prototype font image and the target image, using *best-buddies* sparse correspondences, extracted from the similarity volume, as a robust signal to fit this alignment. Finally, we perform local refinement to optimize the relative positions of each stroke and their internal configurations, when deviation from the canonical configuration is necessary. We describe each step in turn below, with further implementation details provided in the appendix.

### 4.1 SEMANTICALLY-ADAPTED 4D SIMILARITY VOLUME

Our system is based on the guiding assumption that local similarities between regions in the two images can be used to compute a structurally-consistent matching between the prototype sign structure and the target cuneiform sign. As a backbone for computing meaningful similarity scores, we use diffusion features (DIFT Tang et al. (2023)), which leverage the strong geometric and semantic understanding of a generative text-to-image model to represent image features for discriminative tasks. These features are calculated as intermediate activations the model's denoising component, applied to the input image with added random noise. However, standard generative models

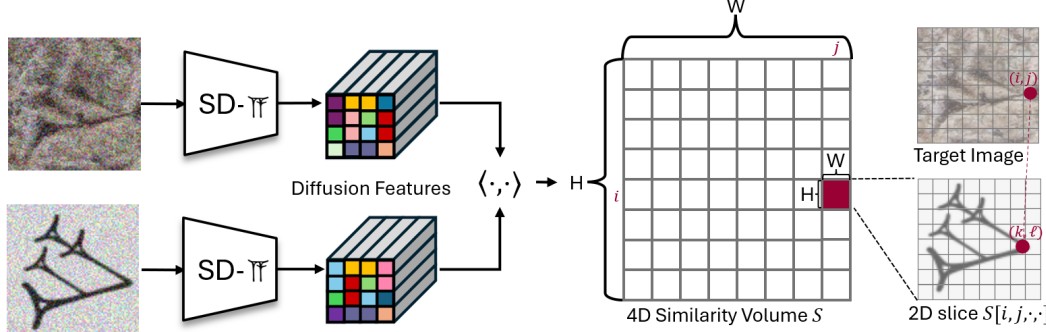

Figure 3: **DIFT-Based Best-Buddies Correspondences**. Noised images are passed through our fine-tuned denoising diffusion model SD-🏶 to extract deep Diffusion Features (DIFT), used to calculate the 4D similarity volume $S$. For each region $(i, j)$ in the target image, we examine the 2D slice $S[i, j, \cdot, \cdot]$, and determine the indices $(k, \ell)$ which maximize its value. Symmetrically, for each region $(k, \ell)$ in the prototype we find the corresponding region in the target by maximizing the 2D slice $S[\cdot, \cdot, k, \ell]$. If these two regions correspond to each other, they are identified as *best buddies*.

are typically pretrained on natural images from the Internet, with cuneiform scans being out-of-distribution. We thus fine-tune the generative model Stable Diffusion (Rombach et al., 2022) on cuneiform image scans, which we indicate as SD-🏶, using the cuneiform sign name as its accompanying text prompt. We use this as our vision backbone for calculating DIFT features.

We apply DIFT with SD-🏶 to our prototype and target images to obtain feature vector maps $F^{(p)} = (\mathbf{f}_{i,j}^{(p)})_{i,j}$ and $F^{(t)} = (\mathbf{f}_{i,j}^{(t)})_{i,j}$ respectively of unit-normalized feature vectors. Each feature map is a $C \times H \times W$ tensor, where $C$ is the number of feature dimensions and $H$ and $W$ are the spatial dimension of the feature map (lower-resolution than the original image, as each feature vector has a larger receptive field in the image). Using these features, we calculate the four-dimensional similarity volume $S = (\mathbf{f}_{i,j}^{(p)} \cdot \mathbf{f}_{k,\ell}^{(t)})_{i,j,k,\ell}$. This $H \times W \times H \times W$ tensor, visualized in Figure 3, contains the pairwise cosine similarities between features encoding patches of the prototype and target images.

## 4.2 GLOBAL ALIGNMENT FROM BEST-BUDDIES CORRESPONDENCES

While $S$ provides a strong similarity measure, it does not encode geometric constraints on the overall matching between the two images. For instance, multiple regions in one image may all have high similarity scores with a single region in the other image, but mapping them all to the same target region will result in a degenerate solution. To robustly identify a sparse set of best-matching pairs of regions, we follow prior work (Oron et al., 2017; Drory et al., 2020) by identifying *best buddies*, defined as pairs of patches in the two images which are mutual nearest-neighbors according to their similarities scores in $S$. Formally, this is defined as pairs of coordinates $(i, j)$ and $(k, \ell)$ such that $(i, j) = \arg\max_{i,j} S_{i,j,k,\ell}$ and $(k, \ell) = \arg\max_{k,\ell} S_{i,j,k,\ell}$. See Figure 3 for an illustration.

Using these sparse correspondences, we fit an affine transformation with least squares estimation, defining a global alignment $G$ of the prototype to the target image. This transformation allows for basic deformations while preserving the overall structure of the prototype. We learn the parameters

$$G = \begin{bmatrix} g_{11} & g_{12} & g_{13} \\ g_{21} & g_{22} & g_{23} \\ 0 & 0 & 1 \end{bmatrix}$$

permitting scaling, rotation, and shear $(g_{11}, g_{12}, g_{21}, g_{22})$ as well as translation $(g_{13}, g_{23})$. This is applied in projective space $\mathbb{P}^2$, i.e. mapping a point $\mathbf{v} = [x, y, 1]^T \in \mathbb{P}^2$ to the point $\mathbf{v}' = G\mathbf{v} = [x', y', 1]^T$.

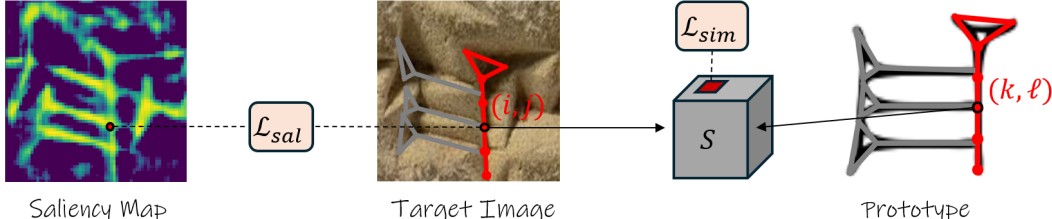

Figure 4: **Local Refinement via Skeleton-Based Optimization.** To adjust the positioning of individual strokes in a sign, our global alignment is followed by a local refinement stage which learns transformations for each stroke. The loss function encourages positioning on salient regions ($\mathcal{L}_{sal}$) while semantically matching the corresponding regions in the prototype image, as measured by feature similarity ($\mathcal{L}_{sim}$). For each stroke (exemplified by the stroke in red above), these objectives are calculated along points sampled from the skeleton (red dots above). The loss also includes a regularization term ($\mathcal{L}_{reg}$) preventing excessive deviation from the global transformation.

To handle outliers in a robust manner, we perform the fitting with RANSAC. For further robustness, we incorporate a prior on inlier points being spread over the majority of the area of prototype and target images, by performing this procedure multiple times (including the stochastic calculation of DIFT features and correspondences from Section 4.1) and selecting the result with the best spread of inlier points across the relevant regions in the images, following Hassner et al. (2014) and as further described in the appendix.

### 4.3 LOCAL REFINEMENT VIA SKELETON-BASED OPTIMIZATION

Our global alignment procedure can roughly align the prototype to the target image. However, as the target image was written by hand, each stroke's location may deviate from the canonical relative position given by the prototype font image. Therefore, we introduce a local refinement stage, illustrated in Figure 4, which allows each stroke to move from the canonical prototype structure and "snap" into place, while avoiding excessive deviations from the global structure representing the sign's identity. We model each stroke's deviation from the global alignment as a projective transformation, allowing for a higher degree of deformation than the affine global transformation. The local transformation of stroke $i$ is parameterized by the matrix

$$P^{(i)} = I + \begin{bmatrix} p^{(i)}_{11} & p^{(i)}_{12} & p^{(i)}_{13} \\ p^{(i)}_{21} & p^{(i)}_{22} & p^{(i)}_{23} \\ p^{(i)}_{31} & p^{(i)}_{32} & 0 \end{bmatrix}$$

where $I$ is the $3 \times 3$ identity matrix. These are applied on top of the global transformation, resulting in per-stroke transformation of the form $P^{(i)}G$. As a projective transformation, this maps a point $\mathbf{v} = [x, y, 1]^T \in \mathbb{P}^2$ to $P^{(i)}G\mathbf{v} = [x', y', z']^T \sim [x'/z', y'/z', 1]^T$. The $p^{(i)}_{jk}$ are all initialized to zero (i.e. each local transformation is initialized as the identity).

We optimize the parameters $p^{(i)}_{jk}$ via gradient descent with the loss function

$$\mathcal{L} = \lambda_{sim}\mathcal{L}_{sim} + \lambda_{sal}\mathcal{L}_{sal} + \lambda_{reg}\mathcal{L}_{reg}$$

where $\lambda_{sim}, \lambda_{sal}, \lambda_{reg}$ are constant weights. We proceed to define each loss term.

**Featural Similarity Loss $\mathcal{L}_{sim}$.** To encourage semantically-correct positioning of strokes, we define a loss to maximize feature similarities between matching points on the prototype and target images under the local transformation. Using the similarity volume $S$ from 4.1, we sample points along the lines connecting skeleton keypoints in the prototype image, calculate their corresponding points under the current global and local transformations, and evaluate their similarity via $S$ with a temperature-weighted softmax applied to each slice of $S$ over the prototype image. This uses differentiable grid sampling to interpolate values of $S$, as $S$ has a lower spatial resolution in comparison to the images.

**Saliency Map Loss $\mathcal{L}_{sal}$.** To encourage the strokes to cover *salient* regions (i.e. areas which appear to contain writing), we calculate a saliency map over the target image and use it to define a

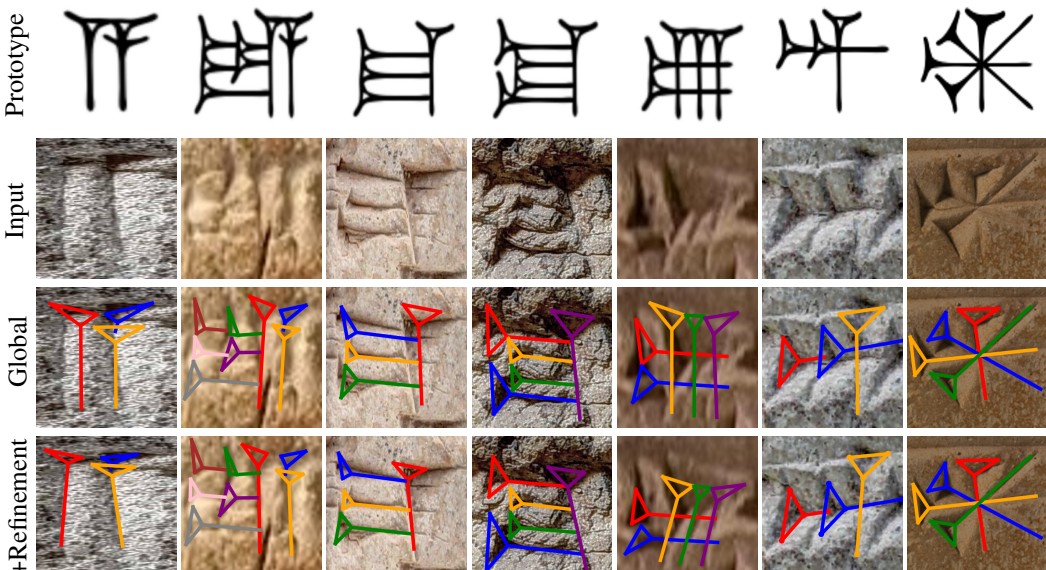

Figure 5: **Qualitative alignment results**, aligning the prototypes (first row) to target cuneiform images (second row). We demonstrate the results after performing global alignment (third row), and the final result after local refinement (fourth row). As illustrated above, the global alignment stage provides a coarse placement of the prototype template, while the refinement stage allows each stroke to slightly diverge from the original prototype, resulting in more accurate alignments.

loss. The saliency map is calculated using the 4D similarity tensor $S$ from Section 4.1; for each region in the target image, we calculate the difference between mean similarities to foreground (black) and background (white) regions in the prototype font image, and post-process by applying adaptive histogram equalization, scaling and setting low values to zero. This yields an approximate segmentation map of the cuneiform sign visible in the scanned image. The loss $\mathcal{L}_{sal}$ is calculated as the mean value of the map sampled at points along the transformed skeleton, selecting points and using temperature-scaled differentiable grid sampling as in Section 4.1.

**Regularization Loss $\mathcal{L}_{reg}$.** To avoid invalid solutions that over-optimize the previous objectives, we add a regularization term that penalizes excessive deformations and solutions that stray from the boundaries of the image. This is defined as $\mathcal{L}_{reg} = \mathcal{L}_{L1} + \mathcal{L}_{oob}$. L1 regularization loss is given by

$$\mathcal{L}_{L1} = \frac{1}{N} \sum_{i,j,k} \left| p_{jk}^{(i)} \right| = \frac{1}{N} \sum_{i} \| P^{(i)} - I \|_{L1}$$

where $N$ is the number of strokes and $I$ is the $3 \times 3$ identity matrix. This penalizes local transformations which greatly deviate from the identity.

The out-of-bounds loss $\mathcal{L}_{oob}$ is defined as zero if all transformed keypoints are within the image boundaries, otherwise as the maximum absolute difference between each transformed coordinate and the image bounds. This models the soft constraint that all keypoints must lie within the image, handling edge cases where the global transformation pushes part of a stroke outside the image bounds.

## 5  EXPERIMENTS

We conduct various experiments, evaluating our method both directly on our benchmark of expert-annotated real photographed signs (Section 5.1) and by testing it on a downstream OCR benchmark (Section 5.2). We also present qualitative results (Figure 5 and appendix), which further highlight the complexity and unique challenges of the task and setting addressed in our work. Finally, we discuss limitations of our approach (Section 5.3). Further implementation details and results are provided in the appendix.

| Method | F1@20 | F1@30 | F1@40 |
|---|---|---|---|
| SIFT (Lowe, 1999) + RANSAC | 2.56% | 3.78% | 5.01% |
| DINOv2 (Oquab et al., 2024) | 12.33% | 21.88% | 31.04% |
| DINOv2 + RANSAC | 16.12% | 28.19% | 37.93% |
| DIFT (Tang et al., 2023) | 16.14% | 25.80% | 33.79% |
| DIFT + RANSAC | 13.13% | 21.96% | 30.15% |
| Ours (w/o refinement) | 21.31% | 37.08% | 50.13% |
| Ours (full) | **27.14%** | **42.09%** | **52.43%** |

Table 1: **Alignment evaluation**, measuring keypoint localization at various distance thresholds. We compare against several generic correspondence matching baselines, including a geometry-based method (SIFT) and deep feature-based methods (DINOv2, DIFT). As illustrated above, our method significantly outperforms these baselines. Furthermore, our local refinement stage provides a performance boost beyond learning simply a global transform.

## 5.1 ALIGNMENT EVALUATION

To evaluate the quality of alignment, we curate a test set of ground-truth (GT) alignments from manual annotations by expert archeologists. Provided with photographs of cuneiform signs from the eBL dataset Cobanoglu et al. (2024), the experts annotated the position of strokes by indicating groups of four keypoints. Images were marked for exclusion from the test set if they were poor quality or show sign variants differing from the corresponding prototype font images. In total, our test set contains 272 images with expert annotations, covering 25 different sign types, which were not seen in training of SD-𒁹. A breakdown of this test set and more details on our annotation procedure are provided in the appendix.

To quantify performance on this benchmark, we compare the predicted and GT positions of stroke keypoints; given a fixed distance threshold, we consider a predicted keypoint as a true positive prediction if it is within the given threshold of a GT keypoint. We report F1 scores for several distance thresholds (along with precision and recall in the appendix).

In Table 1 we compare our solution to generic correspondence matching techniques. We compare against a traditional geometry-based baseline that computes SIFT (Lowe, 1999) features and aligns the images using RANSAC. We also compare against two deep feature-based techniques (DINOv2 (Oquab et al., 2024), DIFT Tang et al. (2023)), applied by assigning each keypoint to the corresponding point of maximum similarity in the target image.

We see that *ProtoSnap* significantly outperforms the baseline methods at localization, as our solution explicitly considers the complex structure of the sign. We also report performance of our approach without local refinement (*i.e* performing the first step of global alignment alone); we see that the final step of local optimization indeed achieves better alignment as reflected in our metrics. This may also be seen visually in Figure 5, illustrating the overall global alignment and more precise results yielded by local refinement. In the appendix, we show an ablation study of the different components of our system (such as using fine-tuned SD-𒁹 and each of our loss terms). These results provide further motivation for the design of our full system. We also provide additional qualitative results.

To further evaluate the practical usability and effectiveness of *ProtoSnap*, we conducted a survey of 12 assyriologists, finding that users are approximately twice as likely to prefer scans with our aligned skeleton overlaid to an overlay without our alignment applied. Further details regarding this user study can be found in the appendix.

## 5.2 LEARNING OCR FROM ALIGNED DATA

We show the downstream benefit of our approach by using *ProtoSnap* alignments to produce structurally-diverse synthetic training data for an OCR system. As cuneiform signs are highly diverse, with number and arrangement of strokes varying significantly between eras, regions, and individuals, a model trained to produce synthetic data conditioned on sign type alone may struggle to depict the correct configurations of signs, particularly rare signs with few to no attestations

| Training Data | Accuracy | | Balanced Accuracy | |
|---|---|---|---|---|
| | All | Rare | All | Rare |
| CSDD⋆ (Dencker et al., 2020) | 58.43% | 25.84% | 34.57% | 16.45% |
| +SD-𒀭 data | 61.56% | 38.56% | 39.02% | 31.13% |
| +CN-𒀭 data | **64.14%** | **53.17%** | **43.57%** | **39.98%** |

Table 2: Sign classification performance when training on previously collected real data (CSDD), and with added data generated using our fine-tuned diffusion model (SD-𒀭) or ControlNet trained using alignments from *ProtoSnap* (CN-𒀭). Our solution demonstrates improved OCR results, with structural augmentations showing increased performance relative to direct unconditional image generation, especially on rare signs (signs that have less than 50 occurrences in the real train set). ⋆ denotes our reproduced model, as further described in the appendix.

in the existing data. Furthermore, conditioning on sign type does not allow to specify the exact variant of the sign, resulting in generation of the most prevalent variant in the data. As we show, conditioning on a skeleton-based structures provides control over the generated fine-grained structures yielding synthetic data which better captures the real configurations. To demonstrate the benefit of our approach for the downstream task of OCR, we align prototype skeletons to a set of cuneiform images from the eBL dataset (Cobanoglu et al., 2024), and fine-tune ControlNet (Zhang et al., 2023) to generate new cuneiform images using such skeletons as a condition, instead of a text prompt. This model, denoted as CN-𒀭, allows us to produce new cuneiform sign images with any input structure, not limited to a list of predefined categories or specific structural configurations. We then use this model to generate synthetic training data, as shown in Figure 6. Further details about the model training are provided in the appendix.

We examine the benefit of this generated data for learning cuneiform sign classification. Dencker et al. (2020) report sign classification performance on the CSDD dataset when training a Resnet18 (He et al., 2016) model with supervision from the CSDD training set alone. We compare this to augmenting the training set with the CN-𒀭-generated synthetic data described above. As an additional baseline, we also compare with using SD-𒀭 (see Section 4.1) generations for augmenting sign types which uses categorical sign names as a textual condition rather than structural conditions. We report classification accuracy following Dencker et al. (2020); as this dataset is highly imbalanced, we also report balanced accuracy and performance on rare sign types (signs with less than 50 occurrences in the real training set). Results are summarized in Table 2. As seen there, augmenting the CSDD dataset with synthetic data significantly improves classification performance, with structurally-conditioned data from CN-𒀭 providing a significant boost over synthetic data from SD-𒀭. These results reflect that structurally-controlled generation with CN-𒀭 guarantees generation of the correct sign variant, while SD-𒀭 struggles to produce correct configurations from the sign category alone, as seen in Figure 6.

## 5.3 LIMITATIONS

We note various limitations of our work (visualized in Figure 7). Our alignment procedure requires a canonical sign image and will still fail if the scan displays a structurally different variant of the sign in question. Additionally, our method may fail under extreme deformations, or on low-quality tablets or scans which cannot be feasibly interpreted. Future work might investigate how to calculate a confidence measure to detect such failure cases.

## 6 CONCLUSION

We have proposed the *ProtoSnap* method for estimating the internal structure of cuneiform signs without any direct supervision, by harnessing pairwise comparisons of deep image features in regions of photographed cuneiform images and skeleton-based prototypes. We have curated expert annotations to provide a new benchmark for this task, and have shown that our method significantly outperforms generic correspondence-based techniques. Beyond its direct application for paleographic analysis, we have also shown our method's utility for the downstream task of OCR, by using aligned skeletons for conditional synthetic data generation.

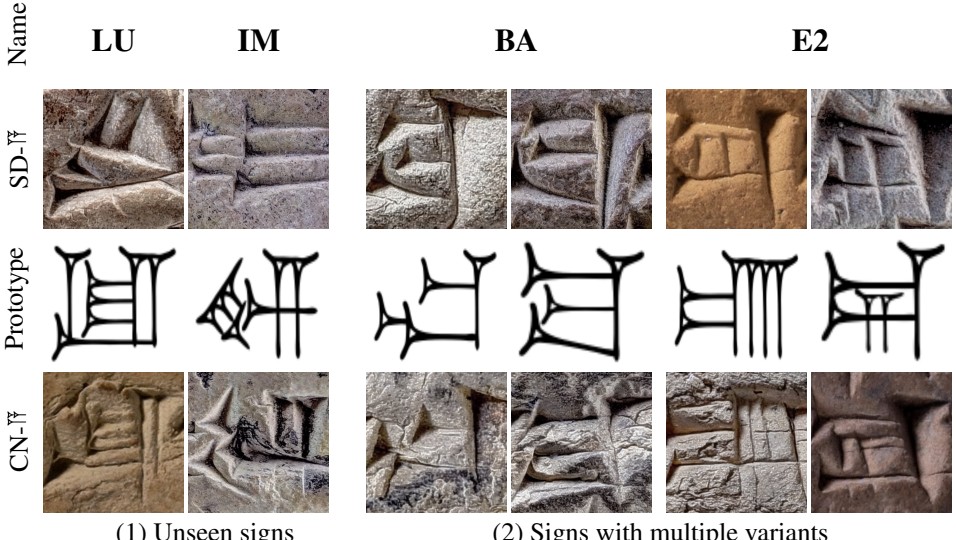

Figure 6: We demonstrate the benefit of producing structurally-controlled synthetic data (denoted as CN-𒀭 above), in comparison to text-conditional generation (denoted as SD-𒀭), using two different scenarios: (1) Text-conditional generation cannot succeed in generating signs unseen during training, while our method can correctly adhere to the structural conditioning provided as input, even for rare or unseen signs. (2) A text-conditional model often generates the sign variation most prevalent during training (*e.g.*, the variants on the right sides above), while our method can generate different variants, yielding a more diverse synthetic set for training downstream models.

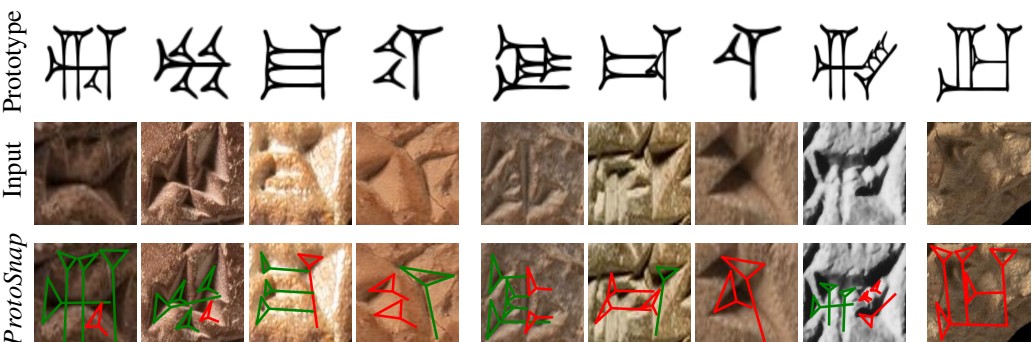

Figure 7: **Limitations of our method**, illustrating examples with significant deformations from the prototype skeleton (left) and structurally different sign variants (middle) and corrupt sign image (right). We visualize correctly-aligned strokes in green, and misaligned strokes in red.

We foresee a range of applications and possible extensions of our work. Our method represents a step towards paleographic analysis on scale, tracking *allographs* (sign variants) for each sign as produced by scribes between time periods, cities, archives, and personal handwriting styles. It could also be used to automate the production of hand copies, currently produced manually by experts to illustrate tablet contents in their original context. While we have focused on the case of single sign scans, we foresee future work extending these results to lines of text, as our method could be incorporated into a pipeline including text detection and localization of individual signs applied either to images or directly to 3D scans of inscriptions. Finally, our per-stroke optimization process is designed for cuneiform signs composed of wedge shapes parametrized by four keypoints, future work might extend this to other ancient scripts with different geometric configurations – for example, by using more flexible primitives such as Bezier splines to model curved lines found in ancient Chinese oracle bone inscriptions. We hope that our work will spur future research on using the internal structure of glyphs in complex scripts such as cuneiform to advance downstream tasks.

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

## APPENDIX

## A EXPERIMENTAL DETAILS

Below we provide additional experimental details. Our code is also provided (zipped in the supplementary material). For all experiments described below, a single A5000 GPU was used. Running the method on a single image takes about 1 minute.

### A.1 IMAGE AND FONT INFORMATION

For all of our tests, we use RGB images with resolution $512 \times 512$, resizing images as needed.

Font images are rendered from the *Santakku* and *SantakkuM* fonts, designed by Sylvie Vanséveren and available on the Hethitologie Portal Mainz. For a uniform appearance, the white margins of the image are cropped, than 10 pixels of white margins are added to each side, and finally the image is resized to $512 \times 512$ resolution.

### A.2 MODEL DETAILS

We use the `CompVis/stable-diffusion-v1-4` checkpoint as our base Stable Diffusion model. We fine-tune this on eBL classification train dataset (Cobanoglu et al., 2024), for 50K iterations with batch size of 4, learning rate of $10^{-5}$ and Adam optimizer. For textual prompts, we use a unique code for each sign type indicated in the eBL dataset. This fine-tuned Stable Diffusion model, SD-𒁹, is used for our DIFT feature calculations as well as in the synthetic data generation for our OCR application. For tests using ControlNet, we fine-tune the base `illyasviel/sd-controlnet-openpose` checkpoint, on 932 samples with their paired alignment created by *ProtoSnap*. We trained it for 20K iteration, with batch size of 4, learning rate of $10^{-5}$ and Adam optimizer. This fine-tuned ControlNet, CN-𒁹, uses another fine-tuned Stable Diffusion model, trained with the same parameters as SD-𒁹, but without a prior from a textual prompt, using the same prompt for all signs: "cuneiform single ancient icon".

For DIFT feature calculations, we average an ensemble of results on four random noises, sampled at timestamp $t = 261$, following Tang et al. (2023). We concatenate features from the second and third upscaling U-Net layers, bilinearly interpolated to the same spatial resolution, yielding a set of $64 \times 64$ feature vectors of dimension 1920.

### A.3 GLOBAL ALIGNMENT DETAILS

To fit our global alignment, we apply RANSAC with 2000 iterations. At each iteration, 5 correspondences are used to fit a least-squares affine transformation, with a distance threshold of 50 pixels used to identify outliers. The transformation with the greatest number of inliers is returned.

As a high-quality set of correspondences should explain the relevant regions in both of the images, we incorporate a prior on inlier points being spread across the images, following Hassner et al. (2014). In particular, we perform the above procedure 8 times, and assign each result a score using the convex hulls of the inlier points in the prototype and scanned cuneiform sign images. For the prototype image, we calculate the proportion $p_{proto}$ of the prototype font foreground contained within the convex hull of inlier points. For the scanned cuneiform sign image, we calculate the proportion $p_{scan}$ of the total area of the image covered by the convex hull of inlier points. Finally, we select the result with maximum score $p_{proto} \cdot p_{scan}$, to encourage the global transformation to be based on matches between regions covering most of the scan and prototype font.

## A.4 LOCAL REFINEMENT DETAILS

Our total loss uses coefficients $\lambda_{sim} = 1.0$, $\lambda_{sal} = 3 \times 10^{-4}$, and $\lambda_{reg} = 10^{-4}$.

To calculate the saliency map used for the saliency loss, we compute differences in mean similarities as described in the main paper, apply CLAHE histogram equalization with clip limit 10.0 and tile grid size $(2, 2)$, set values below the mean to zero, and scale values to the range $[0, 1]$. This yields a scalar field of resolution $64 \times 64$.

For both semantic similarity and saliency losses, at each iteration values are sampled at both each of the keypoints in the skeleton, and at 8 randomly-sampled points along each line segment in the skeleton connecting keypoints. Loss values are averaged over all of these points. The sampled points are sampled uniformly from the lines in the prototype images, and then transformed using the current global and local transformations to obtain corresponding points in the target scanned cuneiform sign image. Loss values are computed with differentiable grid sampling, using bilinear interpolation over scalar fields (the similarity values in respective slices of $S$, and saliency map values) each of which is passed through a softmax with fixed temperature parameter 100.

To perform optimization, we apply gradient descent for 100 iterations with learning rate 0.01 and Adam optimizer, updating the the parameters of the local transformations of all strokes.

## A.5 DATASET DETAILS

Both the training and the test datasets are taken from the eBL classification dataset Cobanoglu et al. (2024) which consists of scanned imaged of cuneiform signs. The dataset was originally split to train and test by source tablet. The train set included 34,868 sign images, representing 362 sign types and was used to train SD-𒁹. The test set was used to select the 272 images for our test set (as described in 5.1), focusing on signs for which we have an available prototype, and the samples in the set match the prototype structure variant. The final test set consists of 272 samples from 25 different signs, not seen in training, varying from 2 strokes per sign to 8.

The full dataset comprises around 40% from the Neo-Babylonian period (1000–600 BC), around 20% from the Neo-Assyrian (1000-609 BCE), and less than 10% from the following periods: Ur III (2100–2002 BCE), Old Babylonian (2002-1595 BCE), Old Assyrian (1950–1850 BCE), Middle Babylonian (1500–1000 BCE), Late Babylonian (600 BC–100 AD), Persian (539-331 BCE), Hellenistic (331-141 BCE), Parthian (141 BCE-100 CE). The dataset represents the Akkadian and Sumerian languages used at those eras.

## A.6 USER SURVEY DETAILS

For the users survey, we have asked experiences assyriologists to review 35 cuneiform signs. For each signs, we presented them with 3 options - a plain sign, a sign with an overlaid prototype without alignment, and a sign with overlaid prototyped aligned using *ProtoSnap*. For each sign we asked the user to select the image in which they can most easily identify the sign. 12 assyriologists had answered the survey. In all but three signs, the users preferred an overlaid option to the plain one, and out of those, the users preferred the aligned option 65% of the time (in 21 signs).

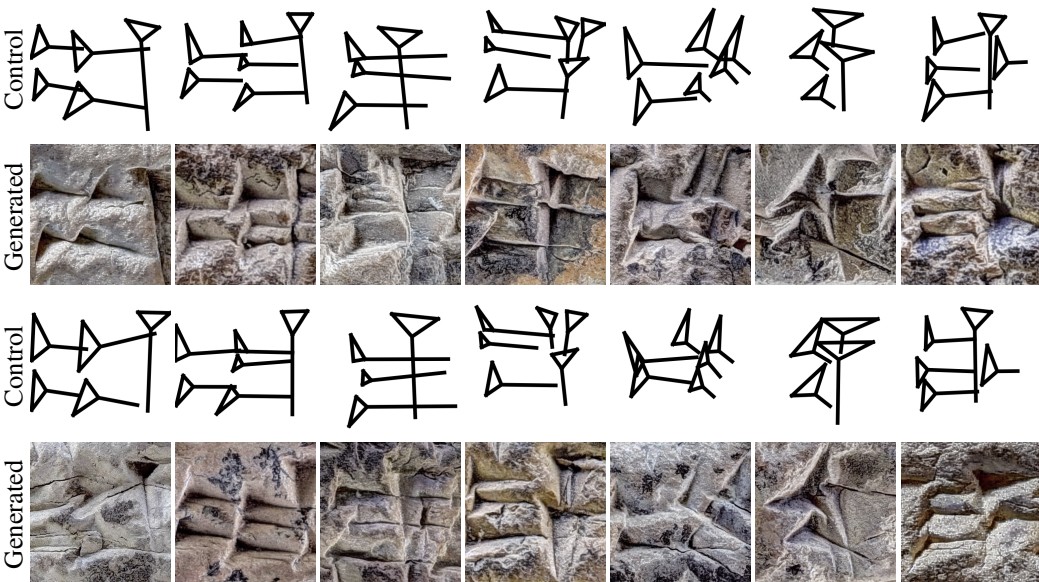

Figure 8: Examples for data generated using our fine-tuned ControlNet model CN-⊭, where the control is an image of a prototype sign (with two different added transformations).

## A.7 OCR EXPERIMENT DETAILS

For the OCR experiment we have generated 50 samples per each sign in the test set (180 signs in total), using our fine-tuned SD-⊭ described above. In addition, we used CN-⊭ to generate 50 samples for each sign we have available prototype (124 signs in total). To better mimic human handwriting, we augmented the skeletons by applying small random transformation on the entire skeleton and on each stroke individually, creating a diverse set of controls for each sign. Figure 8 shows examples of such generated data. Both generations were done using 50 inference steps and classifier-free guidance scale 7.5.

For the experiment we used 6205 samples from the CSDD data (consisting of data from all URLs which were not broken in the dataset), 3299 samples in the train set and 2906 in the test. Training ResNet18 on this dataset alone was done for 10 epochs, using batch size of 64, learning rate of $10^{-3}$ and Adam optimizer. Training with generated data (both from SD-⊭ and CN-⊭) was done for 8 epochs, and then fine-tuned 10 more on real data only, using batch size of 64, learning rate of $5^{-4}$ and Adam.

Note that our reproduced baseline on CSDD achieves slightly higher accuracy than reported by Dencker et al. (2020), but the test dataset includes URLs which are no longer operational which may contribute to this slight discrepancy in results.

## B ADDITIONAL RESULTS

Figure 9 shows additional examples, illustrating *ProtoSnap* applied to various samples of the same sign. Figure 10 shows *ProtoSnap* results on our manually annotated test set, compared to the baseline of applying DIFT directly (assigning each keypoint to the region of maximal feature similarity).

Figure 11 shows *ProtoSnap* results on a new, previously unseen dataset, JOCCH (Rusakov et al., 2020) , which contains signs from the Hittite language, as opposed to the Akkadian and Sumerian from which the training and test set are composed. Those results show that our method is robust and can be generalizable to other usages of the cuneiform writing system.

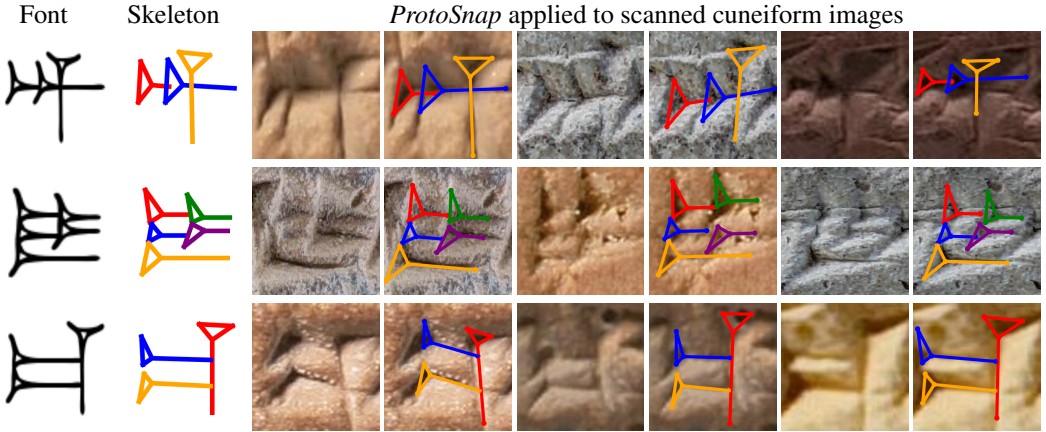

Figure 9: Examples of *ProtoSnap* applied on photographed cuneiform signs of varying structure, illumination conditions and degrees of intactness.

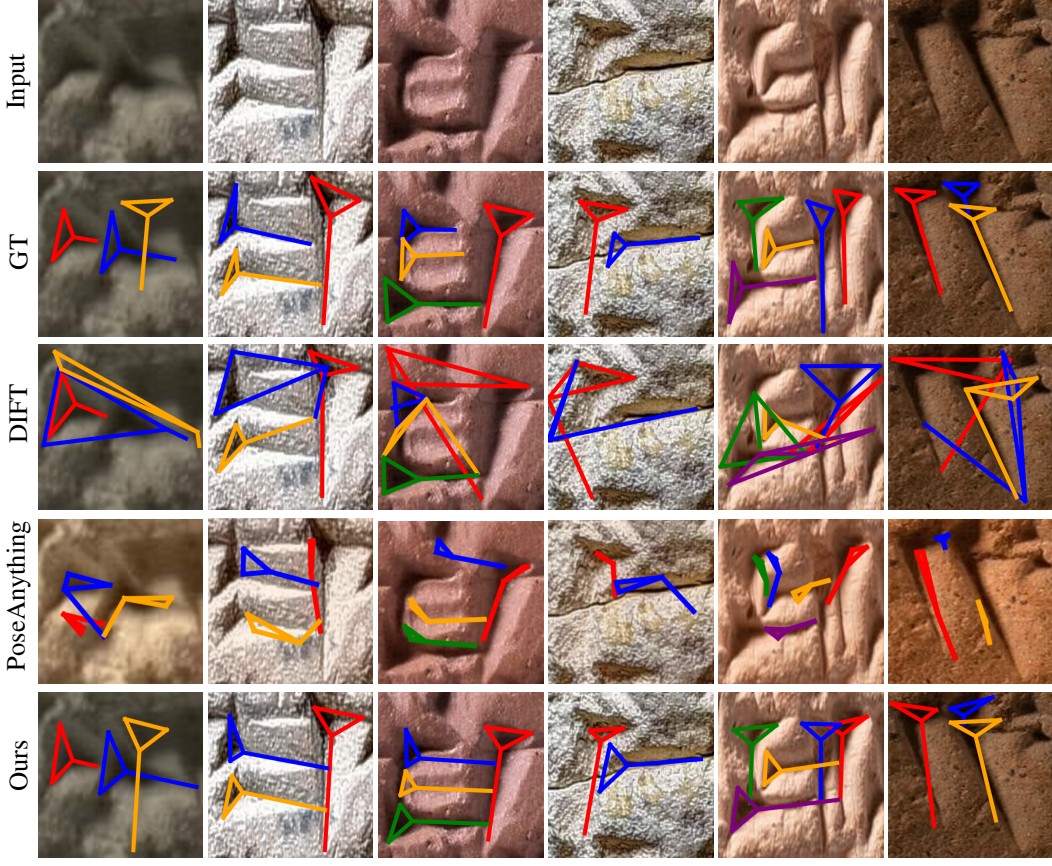

Figure 10: Results of *ProtoSnap* on our manually annotated test set, with DIFT and PoseAnything (Hirschorn and Avidan, 2023) shown for comparison. We can see that our method produces alignments which are much closer to expert annotations and is generally less sensitive to outliers.

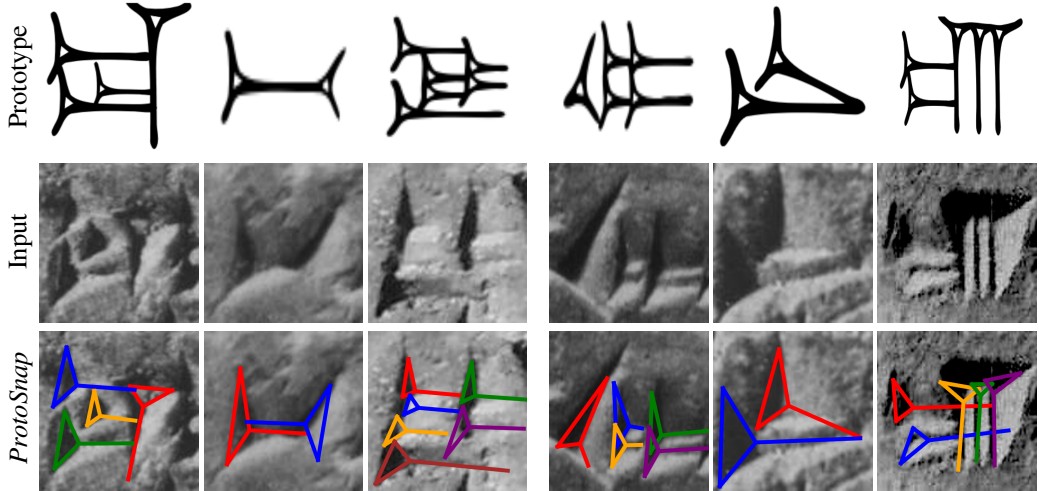

Figure 11: *ProtoSnap* applied on images from a different dataset and language (Hittite), showing that the method is robust and generalizable to various usages of cuneiform writing system. The 3 images on the left show signs from types (names) unseen in the training data, further emphasizing the generalizability of the method.

| Method | threshold = 20 | | | threshold = 40 | | |
|---|---|---|---|---|---|---|
| | Precision | Recall | F1 | Precision | Recall | F1 |
| SIFT (Lowe, 1999) + RANSAC | 2.59% | 2.53% | 2.56% | 5.49% | 4.61% | 4.91% |
| DINOv2 (Oquab et al., 2024) | 12.42% | 12.25% | 12.33% | 32.09% | 30.05% | 31.04% |
| DINOv2 + RANSAC | 16.22% | 16.01% | 16.12% | 38.47% | 37.42% | 37.93% |
| DIFT (Tang et al., 2023) | 16.18% | 16.10% | 16.14% | 34.32% | 33.27% | 33.79% |
| DIFT + RANSAC | 13.15% | 13.11% | 13.13% | 30.43% | 29.88% | 30.15% |
| Ours (w/o refinement) | 21.38% | 21.23% | 21.31% | 50.55% | 49.73% | 50.13% |
| Ours (full) | **27.17%** | **27.10%** | **27.14%** | **52.76%** | **52.10%** | **52.43%** |

Table 3: Precision and recall metrics for the alignment evaluation, on top of F1 metric presented in the main paper, at two distance thresholds.

## B.1 ADDITIONAL METRICS

Table 3 shows precision and recall metrics for the alignment evaluation, on top of F1 metric presented in the main paper. Table 4 show alignment evaluation breakdown per signs, and also provides the number of samples in the test set per sign.

## B.2 ABLATION STUDY

We demonstrate the effect of key parts of our system by ablating them and evaluating performance on our test set. In particular, we ablate the following:

- Use of our fine-tuned SD-𐎶 (rather than base Stable Diffusion)

- Use of *best-buddies* correspondences for computing the global transformation (rather than using all correspondences between prototype image regions and the best-matching regions in the target image according to DIFT)

- Each of the three loss terms in our full loss function

As seen in Table 5, most of these ablations have a significant negative effect on quantitative performance. Removing $\mathcal{L}_{sim}$ slightly improves metrics, but we find this is reflects a qualitative trade-off.

| Sign Name | # Samples | # Strokes | F1@20 | F1@30 | F1@40 |
|---|---|---|---|---|---|
| ME | 20 | 2 | 23.13% | 33.75% | 46.88% |
| A | 19 | 3 | 26.75% | 40.79% | 52.41% |
| IGI | 18 | 3 | 12.50% | 25.23% | 37.50% |
| AN | 19 | 3 | 30.04% | 46.05% | 61.39% |
| UD | 18 | 3 | 16.20% | 29.17% | 44.21% |
| GISH | 19 | 3 | 28.29% | 42.11% | 54.17% |
| MA | 16 | 4 | 26.95% | 44.73% | 57.81% |
| EN | 5 | 5 | 20.00% | 36.00% | 47.49% |
| IR | 17 | 5 | 22.94% | 36.17% | 42.04% |
| IB | 15 | 5 | 35.17% | 55.17% | 63.33% |
| HA | 3 | 5 | 23.33% | 35.81% | 48.33% |
| UR | 16 | 5 | 32.66% | 49.06% | 57.50% |
| RI | 13 | 5 | 29.81% | 44.03% | 57.11% |
| RU | 15 | 5 | 24.67% | 37.50% | 45.83% |
| DIM2 | 2 | 5 | 25.00% | 35.00% | 42.50% |
| DI | 6 | 5 | 24.17% | 43.71% | 51.25% |
| U2 | 9 | 5 | 21.67% | 41.39% | 53.88% |
| DIB | 3 | 6 | 23.61% | 34.01% | 46.52% |
| SA | 3 | 6 | 36.79% | 45.13% | 52.00% |
| GI | 1 | 7 | 28.57% | 46.43% | 53.57% |
| DA | 1 | 7 | 28.57% | 42.86% | 50.00% |
| KA | 12 | 7 | 26.64% | 44.94% | 55.79% |
| ZI | 10 | 7 | 31.43% | 42.50% | 50.35% |
| A2 | 4 | 8 | 29.69% | 44.92% | 55.47% |
| ZE2 | 8 | 8 | 39.45% | 54.49% | 60.35% |

Table 4: **Performance breakdown by sign type**. We report alignment performance of our model over the different annotated signs. We also provide the number of strokes in each sign (#Strokes) and number of samples of the sign in the test set (#Samples).

We foresee an expansion of our test set or development of additional test metrics for this task to better capture this performance.

| | F1@20 | F1@30 | F1@40 |
|---|---|---|---|
| *ProtoSnap* (ours) | 27.14% | 42.09% | 52.43% |
| $-$SD-$\mathcal{F}$ | 19.01% | 31.89% | 41.25% |
| $-$best buddies | 26.76% | 41.76% | 52.76% |
| $-\mathcal{L}_{sim}$ | 27.37% | 42.61% | 53.19% |
| $-\mathcal{L}_{sal}$ | 21.13% | 37.48% | 50.08% |
| $-\mathcal{L}_{reg}$ | 26.72% | 39.66% | 47.41% |

Table 5: Ablation study results, demonstrating differences in performance when removing key parts of our system. Most ablations negatively impact quantitative results, further explained in Section B.2.

## C ANNOTATION DETAILS

Our expert annotations were performed by Assyriologists who participated in this research.

Below, we provide further details on our annotations collected via crowdsourcing, used to annotate keypoints in prototype font images and in scanned cuneiform signs. We then connected the keypoints manually ourselves, creating the prototype skeleton.

## C.1 IRB APPROVAL, PARTICIPANT SOURCING, AND COMPENSATION

Our annotation tasks, approved by our institution's IRB, were conducted on the Amazon Mechanical Turk (MTurk) crowdsourcing platform. We published our tasks for MTurk workers with at least 1000 completed HITs (MTurk tasks) and a HIT approval rate of at least 95%. Workers were compensated $0.25 for each font annotation, corresponding to the duration of this task.

## C.2 ANNOTATION TASK INSTRUCTIONS

*In this task, you will indicate keypoints on ancient character ("cuneiform") to indicate the location of each stroke. Please indicate each stroke with four keypoints as shown here:*

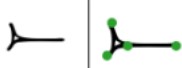

*If there are multiple strokes, please indicate each stroke in a separate color using four keypoints per stroke, as in these examples:*

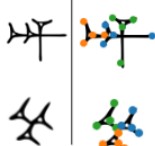

*Make sure the four keypoints are in the locations as shown above – three indicating the corners of the stroke's triangular head, and one indicating the end of its tail.*

*Use the point tool to place points on the requested target(s) of interest: Four each stroke in the character, place four keypoints of the same color, using your mouse to click on each keypoint. Use the four keypoints described in the instructions: for each stroke, three points indicating the corners of the stroke's triangular head, and one indicating the end of its tail. Make sure to indicate every stroke seen in the glyph.*

