# OpenReview forum: "ProtoSnap: Prototype Alignment For Cuneiform Signs"
_ICLR.cc/2025/Conference — ICLR 2025 Poster_

### Official Review · Reviewer_UGZ9 · 2024-10-28

**Soundness:** 4
**Presentation:** 4
**Contribution:** 3
**Rating:** 8
**Confidence:** 4

**Summary:**

The paper presents a method for aligning cuneiform character prototypes to in-the-wild real character images. The prototype consists of an image of the canonical character, as well as an aligned skeleton representation of the character. The method uses deep image features from a finetuned diffusion generation model to measure the patch similarities between the prototype image and the real image. Given the similarity map, the method first applies a global affine transformation to the skeletal representation, such that the feature similarities are maximized. Then the method applies per-stroke projective transformation to the skeleton strokes, to further maximize the alignment of image features. To regularize the alignment optimization, mutual optimal matching between patches, RANSAC for global transform, as well as saliency, identity and boundary constraints for local transform are applied.

The method has been tested on a new benchmarked collected by expert annotators, and showed improved accuracy than baseline matching algorithms. The aligned image-character pairs also allow for finetuning ControlNet, so that new images can be synthesized for training OCR models, which demonstrates the benefits enabled by the method.

Overall, this paper presents a fluent combination of various image processing and registration tools to solve the problem of cuneiform character recognition to a better state.

**Strengths:**

The paper is well written and well illustrated. Technical designs are presented concisely in the main text and discussed in detail in the appendix. Experiments are extensive, using a new benchmark with images labeled by experts and crowdsourcing. Ablation studies are exhaustive and confirmative of the various technical designs. The use of aligned prototypes for generative data synthesis is particularly interesting, by bridging the power of pretrained ControlNet and the parameterized cuneiform images.

**Weaknesses:**

The main weakness is that only cuneiform characters are considered. The authors did not discuss if the same set of techniques used by the method pipeline can be applied to other types of characters, like oracles. It's desirable to at least discuss the possibilities and challenges, e.g. within the different skeleton structures and permitted deformations.

More analysis of the collected and training datasets can be done, to provide the readers with more understanding of the common signs and variations.

**Questions:**

As mentioned above, I hope the authors can discuss the extension and challenges when applying the method to more types of ancient characters. In particular, oracle bone characters have different organizations than cuneiforms, as an oracle bone character not only consists of multiple strokes, but more importantly the strokes are not fixed in numbers and shapes as the cuneiforms do; instead, the strokes are connected into components depicting certain figures, which can vary to a large extent in terms of the number of strokes and nonrigid deformation. To handle such variations, the assumed projective per-stroke transform may not be sufficient. In particular:
1. Are there any components of the current method that would need significant modification?
2. Could the authors provide a brief analysis of how the skeleton structure and deformation models need to be adapted?

Fig.6 can be more specific about "the most prevalent sign variation during training". Which are these variants and how do you know it? Specifically, the authors can provide more insights into the dataset, by the following means:
1. Provide quantitative data on the frequency of different sign variants in their training set, if available.
2. Explain the method used to determine which variants were most prevalent.
3. Include a brief discussion about how this prevalence might impact the model's performance.

---

> ### Author Response · Authors · 2024-11-23
>
> We thank the reviewer for the constructive feedback and comments. We address the reviewer’s request regarding adoption of the method to other ancient scripts in our global response, including a discussion of modifications required to our skeleton-based method in such cases.
>
> Regarding analysis of our dataset’s contents, we refer to our global response for an overall discussion of its diversity and points we will expand upon in our revision.
>
> Regarding sign variants, we refer to our discussion in Section 3 (lines 155-158) and Figure 5 which illustrate how a single sign type (AN, in the case of the figure) may be represented by multiple configurations of wedges, depending on era, geographical region and scribal preferences. While these variants do not have a canonical representation (they are not represented in Unicode or labeled in our dataset), they moderately correlate with sign era, which we quantify directly (added to Section A.5 in the appendix, lines 789-793) .
>
> We clarify that Figure 6 illustrates that a diffusion model trained to generate signs *only from their categorical name* (e.g. AN) will arbitrarily generate one of such variants, most likely that which occurs most often during training. By contrast, our method for training a conditional diffusion model takes the sign’s structure as input, thus disambiguating the desired variant. Our ProtoStap alignment method, used to train this model, does not require determining variant types for signs, as asked by the reviewer; rather, our method *assumes* a scanned sign and prototype reflecting the same sign variant as input, and performs alignment on them (lines 187-197). Moreover, we find this method to be robust to unseen sign types (i.e. not seen during Stable Diffusion fine-tuning), as shown in our global response.

---

### Official Review · Reviewer_KZk1 · 2024-11-02

**Soundness:** 3
**Presentation:** 3
**Contribution:** 3
**Rating:** 6
**Confidence:** 3

**Summary:**

The paper present a method for fine-grained structure retrieval in cuneiform sing images, given the canonical form of the depicted signs. In more detail, the method first calculates a global alignment using diffusion features, best buddies correspondences and RANSAC. Then,  the aligned template sing is refined via skeleton-based optimization. The authors demonstrate sota results in cuneiform sign alignment and recognition. Last, a new dataset of expert-annotated cuneiform sign images and will be released.

**Strengths:**

1. The authors have done a good job presenting their method to a reader unfamiliar with the subject. The paper is well written and the ideas well presented.
2. The method is novel for cuneiform sign alignment, as it adopts a common tactic from pose/keypoint detection problems in the scope of the presented subject.
3. The method achieves sota results in cuneiform sign alignment, although a more detailed comparisons scheme could have been designed (more details in weaknesses 1).
4. The method achieves sota results in cuneiform sign recognition.
5. A new benchmark dataset of cuneiform sign images with expert annotations will be released.

**Weaknesses:**

1. Comparisons in Table 1 are not clear. To my understanding, for DINOv2 and DIFT the authors directly decide keypoints based on feature similarity without solving RANSAC. On the other hand, the authors employ RANSAC for SIFT features and their method (with or without refinement). In my view, the authors should not focus on a single model for feature extraction, but rather experiment with all of them in the same setting (with or without RANSAC) and present their method as a more general method for template alignment for cuneiform signs.

**Questions:**

1. I kindly ask the authors to provide more details and motivation regarding experimental results in Table 1.
2. To my understanding, cuneiform sign data are few if not limited. How far away is the field from data-driven methods that predict accurate keypoints on sign images without known templates? Could your synthetic data be helpful towards this direction?

---

> ### Author Response · Authors · 2024-11-23
>
> We thank the reviewer for the constructive feedback and comments.
>
> Regarding the request to show more experiments with and without RANSAC, we additionally try DIFT and DINO methods with and without RANSAC, finding a minor difference in all resulting metric values. We have added those experiments to Table 1 in the revised paper. We believe that this clarifies the point regarding the comparisons in this table, but if additional details or motivation are unclear, we are happy to address them on request.
>
> Regarding the question about a model that predicts keypoints without a known template, we believe that this is an interesting direction for research, as keypoint-annotated cuneiform data is virtually non-existent to the best of our knowledge (with our annotated dataset being a key contribution to the community, L378-374). Similarly to how our prototype-driven method provides synthetic data to train an OCR model, we foresee our synthetic data being used to train keypoint detection models, and will add an expanded discussion of this point.

---

> > ### Comment · Reviewer_KZk1 · 2024-11-28
> >
> > I thank the authors for their response. Based on their above comment my considerations for this work have been solved.

---

### Official Review · Reviewer_Bpiu · 2024-11-03

**Soundness:** 3
**Presentation:** 3
**Contribution:** 3
**Rating:** 6
**Confidence:** 3

**Summary:**

This paper introduces ProtoSnap, an unsupervised method for aligning the internal structure of cuneiform signs using generative models and prototype images. The approach improves the recognition of cuneiform signs by refining the alignment of skeletal templates to real sign images. This method leverages deep learning and generative modeling to interpret the complex internal configurations of cuneiform signs, enhancing optical character recognition (OCR) accuracy, especially for rare signs.

**Strengths:**

+ The application of unsupervised learning and prototype alignment to cuneiform signs is novel and shows significant potential.

+ The technical approach is sound, utilizing SoTA techniques in image processing and machine learning.

+ The method has clear applications in digital humanities, aiding the decipherment and study of ancient texts.

**Weaknesses:**

1. There’s a potential risk that the method could overfit to the prototypes it has been trained on, especially if those prototypes do not capture the full variability of the signs in the dataset.

2. It would be good if the authors could report on the computational resources required for implementing the ProtoSnap method. Considering that it involves deep learning models and generative processes for aligning prototypes with actual images, understanding the computational demands is crucial.

3. I'd like to hear the authors' opinion on the potential of ProtoSnap to adapt to other ancient scripts, which often present unique challenges in terms of symbol complexity and degradation patterns. This discussion could provide valuable insights into the versatility and scalability of the proposed method beyond cuneiform studies.

**Questions:**

1. Can the authors detail any specific preprocessing steps required to prepare the cuneiform images before applying ProtoSnap?

2. What are the limitations in terms of computational resources, and how scalable is this approach when applied to large datasets of cuneiform texts?

3. How does ProtoSnap handle extremely degraded or incomplete cuneiform signs where the prototype may not initially align well?

---

> ### Author Response · Authors · 2024-11-23
>
> We thank the reviewer for the constructive feedback and comments. We address the reviewer’s request regarding adoption of the method to other ancient scripts, the diversity of signs in our dataset and the concern regarding overfitting in our global response.
>
> Regarding the request to detail the preprocessing, we clarify that we resize images to 512x512 resolution, and convert all images to RGB format for processing. Prototype font images are rendered using the fonts detailed on (lines 724-725), the white margins are cropped, 10 pixels of white margins are added, and finally the image is resized to 512x512 resolution. We have expanded Section A.1 in the appendix (lines 721-727) of the revised paper with the full preprocessing details.
>
> Regarding computational resources required for our method, all experiments use a single A5000 GPU (line 718), used for fine-tuning and extracting diffusion features, and for test-time optimization. Running the method on a single image takes about 1 minute. By batching images, this can be parallelized for efficient inference on a large dataset. We have added these details to the appendix (lines 718-719).
>
> Regarding poor-quality images, we note this as a limitation of our method in Section 5.3. We have updated Figure 7 to include this case, illustrating that this may result in poor output alignments.

---

### Official Review · Reviewer_xssh · 2024-11-06

**Soundness:** 3
**Presentation:** 2
**Contribution:** 2
**Rating:** 5
**Confidence:** 4

**Summary:**

This paper proposes a novel approach to handle the complex internal structure of cuneiform signs, called ProtoSnap, an unsupervised method that utilizes deep generative models and prototype font images to estimate the fine-grained internal structure of cuneiform signs.

**Strengths:**

Originality: ProtoSnap's use of deep diffusion features and skeleton - based prototypes for unsupervised cuneiform sign alignment is novel.

Quality: The overall flow of the methodology section is sound and logical.

Clarity: This paper is clearer on the whole, from the introduction part of the cuneiform research background and the limitations of the existing methods, which naturally leads to the research objective, i.e., to propose the ProtoSnap method to solve the problem of analysing the internal structure of the cuneiform symbols.

Significance: This work was instrumental in the development of the field of cuneiform writing.

**Weaknesses:**

1. While this paper presents a new benchmark for evaluation, the current dataset may not be fully representative of the variety of cuneiform symbol variants and writing conditions present in the historical record.
2. The superiority of the method proposed in this paper is not reflected in the related work.
3. 4D similarity volumes in section 4.1 are not clearly described.
4. While the method shows promise for cuneiform signs, its adaptation to other ancient writing systems or complex symbol sets may not be straightforward.
5. There are too few comparative experiments to adequately demonstrate the superiority of the proposed method.

**Questions:**

1. "This H ×W×H ×W tensor, visualized in Figure 3, contains the pairwise cosine similarities between features encoding patches of the prototype and target images." I'm confused about the H ×W ×H ×W.

2. The proposal includes a user study to assess the practical usability and effectiveness of the ProtoSnap methodology from the perspective of an end user such as an archaeologist or historian.

3. Recommendations for fuller experiments.

---

> ### Author Response · Authors · 2024-11-23
>
> We thank the reviewer for the constructive feedback and comments. We address the reviewer’s concerns regarding adoption of the method to other ancient scripts and the diversity and coverage of our dataset in our global response.
>
> Regarding the 4D similarity volume, we clarify that H and W refer to the dimensions of the input images (the prototype and target image). Each entry in the volume, $V[i,j,k,l]$, consists of the cosine similarity of the pixel $(i,j)$ in the prototype image and the pixel $(k,l)$ in the target image, as defined on L236. We hope this clarifies the confusion and will happily add further clarifications to the paper upon request.
>
> Regarding comparative experiments, we refer to Tables 1–2, where we compare our method to standard SOTA correspondence matching methods (e.g. DIFT) and SOTA cuneiform OCR. We refer to our response to KZk1 performing additional comparisons with RANSAC applied to these feature matching methods (added to Table 1). Furthermore, addressing the reviewer’s concern about methods in our related work, we added a qualitative comparison to _PoseAnything_, a previous work mentioned in related work. The results were added to figure 10 in the Appendix (lines 881-885) showing that the method fails on cuneiform.
>
> We thank the reviewer for the proposal of a user study to assess the usability of the method. To this end, we conduct a survey of 12 assyriologists, finding that users are approximately twice as likely to prefer scans with our aligned skeleton overlaid to an overlay without our alignment applied. We will add further details in our revision

---

### Author Response · Authors · 2024-11-23
**We respond here to shared concerns regarding adoption of the method to other ancient scripts and regarding diversity of our dataset.**

We thank the reviewers for their constructive comments. We respond here to shared concerns regarding adoption of the method to other ancient scripts and regarding diversity of our dataset.

**Applicability to other ancient languages and scripts:** (xssh, Bpiu, UGZ9)

Our work follows a line of works applying machine learning specifically to cuneiform (detailed in the related works section, lines 108-122); these focus on cuneiform due its historical significance, its unique visual and structural characteristics as three-dimensional indentations in clay under varying lighting conditions (line 111), and diverse variations in canonical sign shapes over time and geographic region (line 112). As described in Sections 3–4, our method assumes signs are composed of wedges indicated by four keypoints, a structural assumption valid for cuneiform but not directly applicable to other writing systems.

Nevertheless, we note that our work is applicable to a number of ancient languages, and we believe it also bears relevance for future work on additional ancient writing systems.

Cuneiform was used as a writing system for a variety of ancient languages, spanning various unrelated language families (such as the Semitic language Akkadian, the Indo-European languages Hittite and Old Persian, and the language isolate Sumerian; Radner and Robson 2011). Our method is agnostic to the language represented by cuneiform writing; to demonstrate this, we apply our method on a new dataset to showcase its performance on an additional language (Hittite) which was not seen in our training and original test sets (representing Akkadian and Sumerian texts).  Results were added to the appendix (figure 11, lines 897-917), demonstrating our method's applicability to diverse languages attested in cuneiform.

With this in mind, we believe our method may inspire future work on additional ancient scripts, such as the oracle bone script mentioned by reviewers. While our per-stroke optimization process assumes characters are composed of wedge shapes parametrized by four keypoints, future work might parametrize sign components (such as the radicals used in oracle bone script, and additional East Asian scripts) with more flexible primitives such as Bezier curves. We will add an expanded discussion of these points in our revision.

**Diversity and coverage of signs in dataset:** (xssh, Bpiu, UGZ9)

We address reviewer concerns about the variety of cuneiform signs represented in our dataset, and whether our model may overfit to signs seen in training.

Regarding data variation, we have added an expanded discussion (revised appendix, section A.5, lines 779-791), noting that our dataset covers a wide time range (from 2100 BCE to 100 AD), multiple languages (Akkadian and Sumerian), and wide geographic regions (spanning most of the ancient Near East, reaching modern Turkey, Egypt and Iran). We also note a variety of complexity levels of signs (as detailed in Table 4 breaking down the distribution of signs with different amounts of strokes), and provide additional evaluation results on an additional dataset and language (Hittite; figure 11, lines 897-917). In our answer to UGZ9, we further describe different sign variants present in our dataset.

We address the concern of reviewer Bpiu that we might be overfitting to specific prototypes by noting that our model was trained on scanned cuneiform signs (added clarification to line 783 in the revised paper) while prototype images are only used in our test-time optimization method (line 187). We empirically find that our model generalizes to these prototype images despite not seeing them at train time.

Furthermore, we address a potential concern regarding generalization to sign types not seen by training, by additionally modifying our train set (used to fine-tune the diffusion model for feature extraction) to remove all sign types present in our test benchmark, and re-evaluating our method on our test set. We find that all metrics change by less than 1%, illustrating that our method generalizes robustly to unseen sign types (and hence is not overfit to specific signs). We will update all results with this train-test split in our revision.

---

### Meta-Review · Area_Chair_KwiT · 2024-12-17

**Metareview:**

This work proposes ProtoSnap, an unsupervised approach to extract the visual configuration of cuneiform signs producing an automatic digital hand copy of them. It uses a fine-tuned generative model as a prior on the appearance of cuneiform images to localize the constituent strokes in real cuneiform images. In practice, ProtoSnap leverages deep diffusion features to match skeleton-based prototypes to target cuneiform signs, estimating its structure without labelled examples of real photographed signs. A new benchmark of expert annotations is also provided and the method is evaluated on this task.

This paper received slightly positive evaluations (6 and 8) and slightly negative ones (5 and 5), and after rebuttal the situation leaned to more positive (6, 6, 8, and 5).

Also given the peculiarity of this work, major concerns regarded its applicability to other ancient languages and scripts, the diversity and coverage of signs in dataset, potentially leading to overfitting, weak comparisons with former work and in the experimental analysis, and some other issues about the experiments, request of clarifications of some parts of the methodology, and the request of the computational load.

Authors provided a rebuttal for each of these comments, which proved to be satisfactory for most of the reviewers.
Reviewer xssh, who remained below threshold, did not react after rebuttal, but the answers provided seem reasonable to the AC opinion.

For these reasons, this paper is considered acceptable for publication in ICLR 2025.

**Additional Comments On Reviewer Discussion:**

See above.

---

### Decision · Program_Chairs · 2025-01-22

Accept (Poster)